# Anatomical Considerations When Treating Compensatory Hypertrophy of the Upper Part of the Masseter after Long-Term Botulinum Neurotoxin Type A Injections

**DOI:** 10.3390/toxins12030202

**Published:** 2020-03-22

**Authors:** Kyu-Lim Lee, Hyun Jin Cho, Hyungkyu Bae, Hyun Jin Park, Min Sun Park, Hee-Jin Kim

**Affiliations:** 1Division in Anatomy and Developmental Biology, Department of Oral Biology, Human Identification Research Institute, BK21 PLUS Project, Yonsei University College of Dentistry, 50-1 Yonsei-ro, Seodaemun-gu, Seoul 03722, Korea; kyulimlee@yuhs.ac (K.-L.L.); hkbae410@yuhs.ac (H.B.); hjpark321@yuhs.ac (H.J.P.); 2Labella clinic, 455 Gangnam-daero, Seocho-gu, Seoul 06611, Korea; ogxyy@hanmail.net; 3Clinical Practitioner, 200 W 60th St, New York, 10023 NY, USA; minsunparkdds@gmail.com; 4Department of Materials Science & Engineering, College of Engineering, Yonsei University, 50 Yonsei-ro, Seodaemun-gu, Seoul 03722, Korea

**Keywords:** superficial part of masseter muscle, compensatory hypertrophy, botulinum neurotoxin type A injection, masseteric hypertrophy treatment

## Abstract

The masseter is the most targeted muscle when treating hypertrophy to produce a smooth face shape. Compensatory hypertrophy is a well known clinical sequela that occurs in botulinum neurotoxin (BoNT) treatments and is limited to the lower part of the masseter. Based on the masseteric hypertrophy procedure, which targets a confined area, we predicted the possibility of compensatory hypertrophy occurring in the upper part of the masseter. If the patient complains about an unexpected result, additional injections must be performed, but the involved anatomical structures have not been revealed yet. The aim of this study was to identify the morphological patterns of the masseter. Deep tendons were observed in most specimens of the upper part of the masseter and mostly appeared in a continuous pattern (69.7%). The superficial and deep tendons could be classified into a simply connected form and forms surrounding part of the muscle. In 45.5% of cases there were tendon capsules that completely enclosed the muscle, which can interfere with how the injected toxin spreads. Interdigitation patterns in which the tendons could be identified independently between the muscles were present in 9.1% of cases. The present findings provide anatomical knowledge for use when injecting BoNT into the masseter.

## 1. Introduction

The masseter is a powerful masticatory muscle involved in closing the jaw [1,2]. This muscle has been reported to be more developed in Asians than in Caucasians, and its hypertrophy is typically alleviated by performing botulinum neurotoxin (BoNT) injections [3,4]. BoNT treatment can reduce the volume of the muscle to smooth the face shape when the masseter appears too prominent [5,6,7,8,9,10].

The most common outcomes after injecting BoNT into the lower part of the masseter are compensatory hypertrophy and masseteric bulging. Most patients receiving aesthetic BoNT treatments experience compensatory and stress hypertrophy symptoms [11], along with general localized pain accompanied by function failure [12]. Compensatory hypertrophy has recently been reported to occur not only in the temporalis but also in the upper part of the masseter (Figure 1). These symptoms occur because the roles of other muscles have been partially performed by the masseter, and BoNT injections weaken the muscles to impair their functions [5,13,14]. These clinical complications can also result from uneven BoNT injections into a large area and multiple layers of a muscle.

Several structures around the masseter should be considered when applying treatments where BoNT has already been injected. The zygomaticus major and zygomaticus minor muscles do not need to be considered as important during BoNT treatments of the masseter, if the treating physician completes the procedure based on an accurate anatomical knowledge of its location and depth. In certain cases, the risorius originates from the surface of the masseter, but previous studies have shown that this muscle is limited to within the anterior area [15].

The lower part of the masseter has previously been suggested as optimal for BoNT treatments, because it constitutes the largest part of the muscle belly and this approach prevents the parotid gland from being targeted. Moreover, the cheilion–tragus line was excluded due to the risk of targeting the parotid duct [15,16]. The infusion of BoNT into the parotid gland is associated with the risk of xerostomia, but several studies have verified that this is only a low risk [6]. Furthermore, various cosmetic procedures specifically targeting parotid glands have recently increased in popularity [17]. These studies have demonstrated that the parotid gland is responsible for a small proportion of the total saliva produced, with most of the saliva being secreted by the submandibular gland, which is not related to the symptoms of dry mouth [18]. Recent studies have shown that the anatomical structures of the muscle belly and tendon shapes of the masseter are possible causes of these complications [1].

Many previous studies have attempted to improve the effectiveness and safety of injection treatments for masseteric hypertrophy [15,16,19], but no study has focused on the upper part of the masseter. Therefore, the aim of this study was to provide guidelines for safer and more effective injections by determining the detailed anatomical structure of the upper part of the masseter.

## 2. Results

The specific type with only a superficial tendon was observed in 21.2% (n=7) of the 33 specimens. The patterns of the muscle belly and the tendon structures did not differ significantly in the deep layers of the masseter. The cases identified with deep tendons were categorized into the following four patterns based on their relationships with the superficial tendons:Type 1. Continuous patterns○1a. Simply continuous pattern, in which the deep and the superficial tendons of the masseter are connected and surround the muscle belly.○1b. One-unit capsule pattern, in which the deep tendons completely enclose the parts of the muscle belly as a single capsule.○1c. Two-unit capsules pattern, in which the deep tendons completely enclose the parts of the muscle belly as two capsules.Type 2. Interdigitation pattern, in which the deep tendons are located sporadically in the muscle belly.

### Deep Tendons of the Upper Part of the Masseter

Deep tendons could be distinguished in 78.8% (*n* = 26) of the 33 cases, and they were either connected to the superficial tendons or surrounded by the muscle belly (Table 1).

Type 1, in which there is a morphological relationship between the deep and the superficial tendons, c 69.7% of cases. Types 1a constituted 24.2% (*n* = 8) of the 33 cases (Figure 2), while Types 1b and 1c constituted 30.3% (*n* = 10) and 15.2% (*n* = 5) of cases, respectively (Figure 3). In other words, in 45.1% of cases the deep tendons completely enclosed the part of the muscle belly (Figure 3). In addition, in Type 2, there were sporadic observations of deep tendons, which was observed in 9.1% (*n* = 3) of cases (Figure 4). The classification of the deep tendon pattern of the superficial part of the masseter is shown in Figure 5.

## 3. Discussion

Compensatory hypertrophy is a well known long-term clinical sequela that occurs in BoNT treatments that is limited to the lower part of the masseter. When the functioning of certain muscles is impaired, it is natural that the surrounding muscles increase in strength, possibly resulting in hypertrophy. The temporalis has been thought to be the only muscle affected by a compensatory hypertrophy of the masseter [20,21]. Considering the injection procedure applied for masseteric hypertrophy, which targets a confined area, we can now predict the possibility of compensatory hypertrophy in the upper part of the masseter. This symptom is readily found in patients, especially in those who are receiving long-term repeated BoNT treatments. Temporalis hypertrophy is fortunately usually covered by the hair line, but masseteric hypertrophy is more prominent within the zygomatic bone and can represent an aesthetic problem for patients who are not expecting such side effects.

Anatomical textbooks describe the masseter as a muscle composed of three layers. The superficial part is clearly visible from the surface and contains the masseteric nerve, which is a branch of the mandibular nerve that is divided into middle and deep layers [18].

Previous studies have described the nerve distribution in the masseter. They reported that the nerve branches innervating the deep and the middle layers of the masseter are a few perforator nerves and branches from the anteroinferior nerve group, being mainly confined and distributed within the lower middle third, and anterior third area of the masseter. The perforator branches also supply the superficial layer of the masseter. In addition, the masseteric nerve branches are confined mostly to the lower middle third area, which is concordant with the BTX injection point that is currently being used clinically. Therefore, they strongly recommend the lower middle third area as the most efficient and safe BTX injection area for the treatment of masseteric hypertrophy [19] (Figure 6).

The location of the nerve endings is very important for the effective action of the botulinum toxin. However, the clinical approach requires the consideration of other anatomical structures that may be affected together such as the parotid glands, parotid ducts, risorius muscle and facial nerves. Various guidelines were identified through previous research. Therefore, the injection method that is currently being used, which is based on the cheilion to tragus or to lobule. Nevertheless, the following long-term treatment has been reported for compensatory hypertrophy (Figure 7). We determined that the pattern of the tendon could be one of the causes. The specific mechanism by which BoNT-A relieves tendon structure has not yet been clearly identified, although previous studies of BoNT injection procedures based on pharmacological evidence have indicated that muscle and tendon structures should be considered when attempting to prevent the toxin from spreading [1,22,23].

Most of the superficial region consists of muscle belly, but complex tendon structures are also observed; these tendons subdivide the masseter into multiple layers [1,24,25]. Various studies have investigated the structure of the masseter. Cioffi et al. reported that there are more aponeuroses in the deeper part of the masseter [26]. Lee et al. investigated the deep inferior tendon (DIT) located in the superficial part of the masseter. The DIT has been identified as contributing to paradoxical masseteric bulging [1]. The findings of these studies highlight the importance of having accurate anatomical knowledge to ensure procedural safety. Our study focused on the deep tendon due to its clinical significance and the fact it can be observed in various forms depending on the anatomy of individual muscles. If the DIT is observed in the lower part of the masseter, it is assumed that its structure will be similar in the upper part.

BoNT acts as a muscular tension reliever, because the transcytosis and the retrograde transport of BoNT suppress the diffusion of neurotransmitters across the peripheral nerve. Since BoNT-A acts on nerve endings, an extensive and accurate anatomical understanding of the nerve endings of the targeted muscle is critical for obtaining maximum relief with the minimum concentration of BoNT. Invasive anatomical procedures are of limited use when attempting to find effective BoNT injection sites due to the risk of damaging the muscle and the target nerve endings [22,23].

One previous study suggested that BoNT should be injected under the zygomatic arch and at the mandibular angle to ensure that the toxin spreads evenly [12]. However, the specific anatomical structures considered during the procedure were not reported. This procedure is generally not recommended since subzygomatic depression might occur in certain patients when targeting the upper part of the masseter. However, if the patient complains about compensatory hypertrophy of the upper part of the masseter, additional toxin injections must be performed to produce a uniform and harmonious facial contour. No morphological studies of the upper part of the masseter have been performed, but such studies are needed due to the variety of BoNT injection methods that require a more detailed understanding of the relevant anatomical structures.

Masseteric hypertrophy treatment is a well known procedure targeting the lower third area of the masseter [6,24]. BoNT treatments are now relatively reliable because they have been performed for a long time and with various guidelines for effective and safe procedures being proposed. Nevertheless, patients continue to experience the side effect of subzygomatic depression.

This study was conducted based on the concerns of patients showing side effects of hypertrophy of the upper part of the masseter. Deep tendons were found in the upper part of the masseter of most specimens (78.8%), and they were either related to superficial tendons or located independently. The deep tendons were also observed in a continuous pattern in most specimens (69.7%). Furthermore, the superficial and the deep tendons could be classified into a simply connected form, or the forms surrounding the part of the muscle belly. The most important result was that in 45.5% of cases there were capsules that completely enclosed the muscle belly. The presence of these tendon capsules can interfere with the spread of the injected toxin, so these different patterns need to be considered as important anatomical structures when treating masseteric hypertrophy using BoNT. Interdigitation patterns in which the deep tendons could be identified independently between the muscles were present in only 9.1% of cases.

We identified several structures around the masseter that should have been considered in the non-invasive treatments of previous studies. Among them, in this study, we focused on the situations where tendon structure should be considered. The tendon structures are particularly important for the additional injection of the compensatory hypertrophy treatment, but clinicians should consider this structure when they approach all kinds of non-invasive treatment of the masseter.

In conclusion, the most important consideration of this study is that the additional injection of BoNT should be performed in consideration of the tendon structure, in case of an unexpected result due to long-term treatment. In this procedure, ultrasonography-guided injection procedures could be utilized to achieve more precise treatment protocols that reduce the likelihood of side effects during BoNT injections into the masseter.

## 4. Materials and Methods

Thirty-three hemifaces from 25 Korean and 8 Thai embalmed cadavers (15 males, 18 females; 19 right, 14 left; age range, 55–97 years; mean age, 80 years) were used in this study to identify morphological patterns between the tendon structure and the muscle belly. The current study was performed in accordance with the principles outlined in the Declaration of Helsinki. Appropriate consent from donation process of Yonsei University College of Medicine and approval were obtained from the volunteers and relatives of the cadavers before the dissections were performed.

The masseter was exposed by removing the skin and subcutaneous tissues around the midfacial area along with the superficial musculoaponeurotic system and the parotid gland. The superficial parts of the masseter were then dissected layer by layer, starting from the lower margin of the zygomatic arch, and then retracted to observe the deep tendon structures. The superficial tendons of the masseter were longitudinally incised, retracted anteriorly and posteriorly, followed by the detailed dissection of the internal structures. The continuous tendon around the anterior border of the masseter and its morphological relationship with the muscle belly were observed. Sections of the upper quarter of the masseter were then cut and cross-section images were observed. The shapes of and the relationships between the deep and the superficial tendons of the masseter were observed and classified into different types.

## Figures and Tables

**Figure 1 toxins-12-00202-f001:**
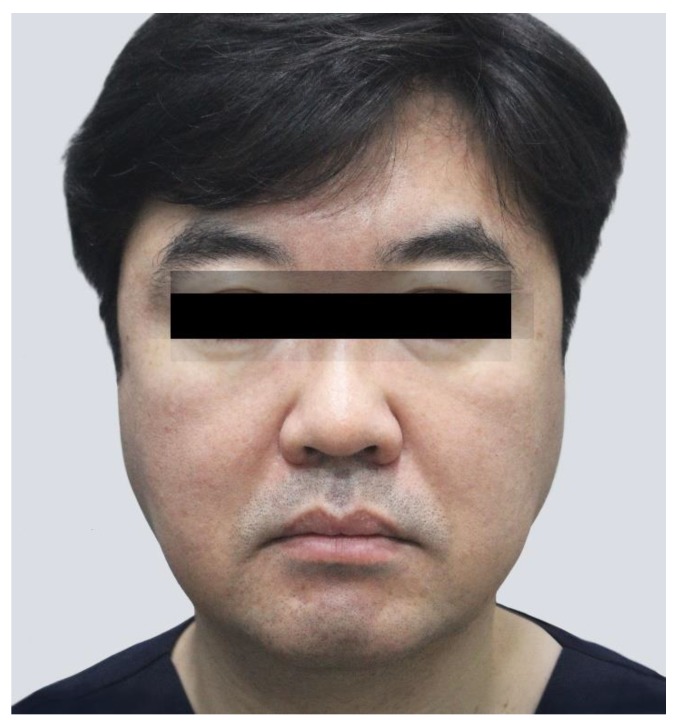
Photography of compensatory hypertrophy of the upper part of the masseter. (Reproduced with the permission of courtesy from Cho HJ, Labella clinic).

**Figure 2 toxins-12-00202-f002:**
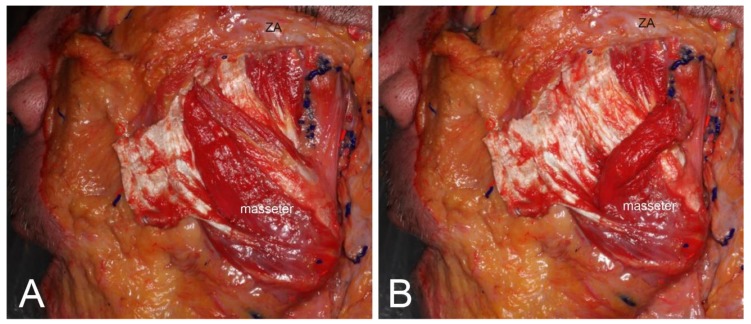
Type 1a, simply continuous pattern. (**A**) The superficial tendon of the surface of the masseter was anteriorly retracted. (**B**) Demonstrating the muscle belly, that is connected and surrounded by the tendons. These tendons are the deep tendon and the superficial tendon of the masseter muscle. ZA, Zygomatic arch.

**Figure 3 toxins-12-00202-f003:**
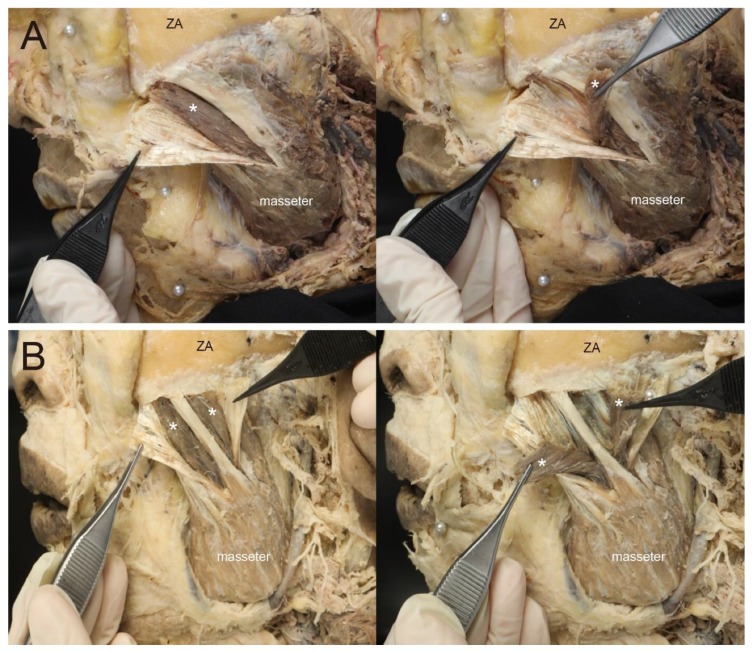
Type 1b and 1c, one-unit and two-unit capsule patterns, respectively. The capsule pattern was subdivided according to the number of capsules. (**A**) One-unit pattern, the superficial tendon was retracted and exposed the muscle belly located inside (muscle belly; white asterisk) (Left); the muscle belly that was surrounded by tendons was separated and indicated (Right). (**B**) Two-unit pattern, the superficial tendon was preserved in between two capsules and retracted apart from both sides of the preserved tendon (Left); The muscle belly was separated and indicated (Right). ZA, Zygomatic arch.

**Figure 4 toxins-12-00202-f004:**
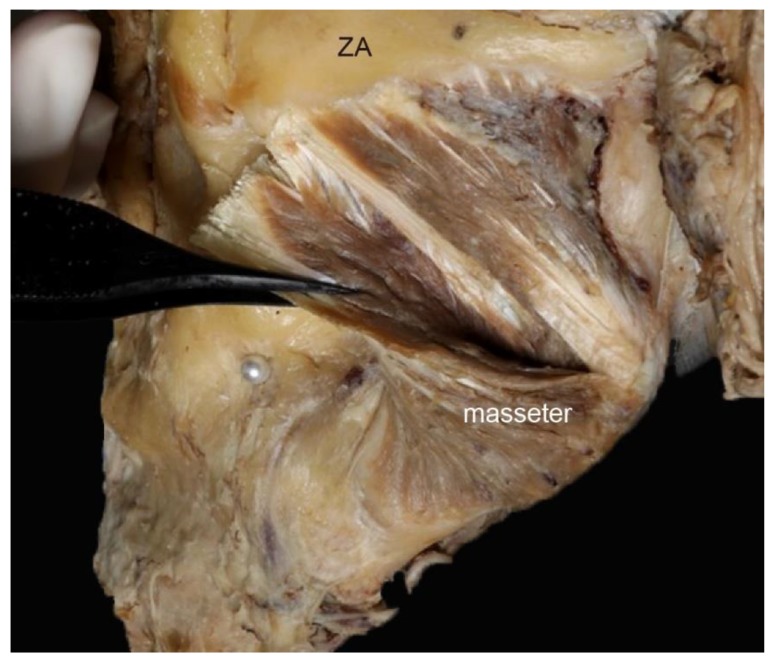
Type 2, interdigitation pattern. The superficial tendon and part of the muscle belly of the masseter were retracted from a lateral to a medial direction. The deep tendons were located sporadically in the muscle belly. ZA, Zygomatic arch.

**Figure 5 toxins-12-00202-f005:**
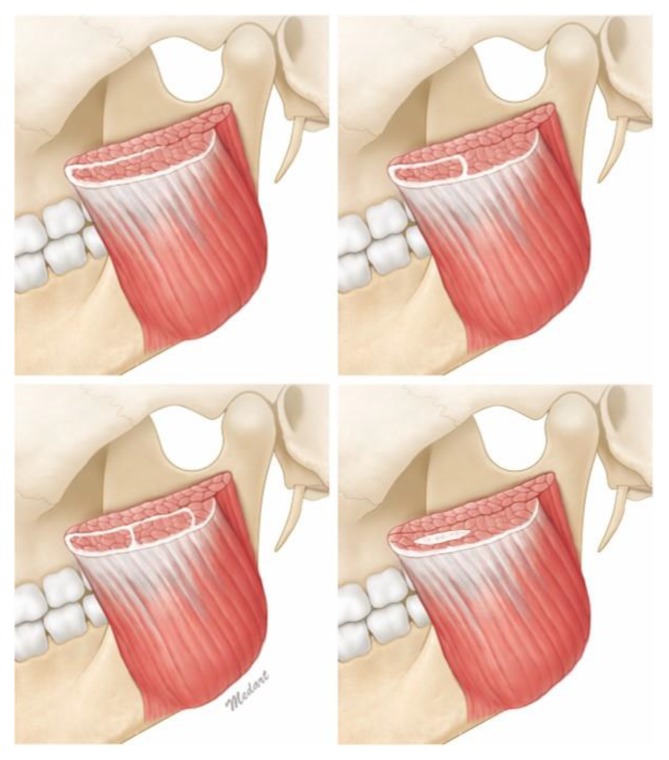
Classification of the deep tendon structure in the superficial part of the masseter muscle. Top left, Type 1a, a simply continuous pattern, in which the deep and the superficial tendons of the masseter are connected and surround the muscle belly; top right, Type 1b. One-unit capsule pattern, in which the deep tendons completely enclose the part of the muscle belly to be a single capsule; bottom left, Type 1c. Two-unit capsules pattern, in which the deep tendons are completely enclosing the parts of the muscle belly to be two capsules; bottom right, Type 2. Interdigitation pattern, in which the deep tendons are located sporadically in the muscle belly.

**Figure 6 toxins-12-00202-f006:**
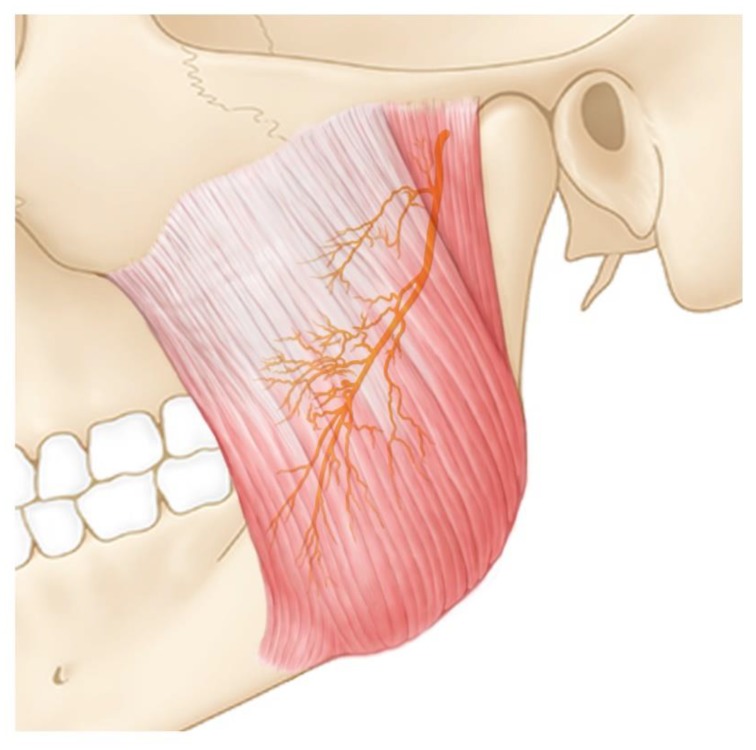
Redrawn illustration on the previous results (Kim et al., Intramuscular Nerve Distribution of the Masseter Muscle as a Basis for Botulinum Toxin Injection. J Craniofac Surg. 2010, 21(2), 588–91) [19].

**Figure 7 toxins-12-00202-f007:**
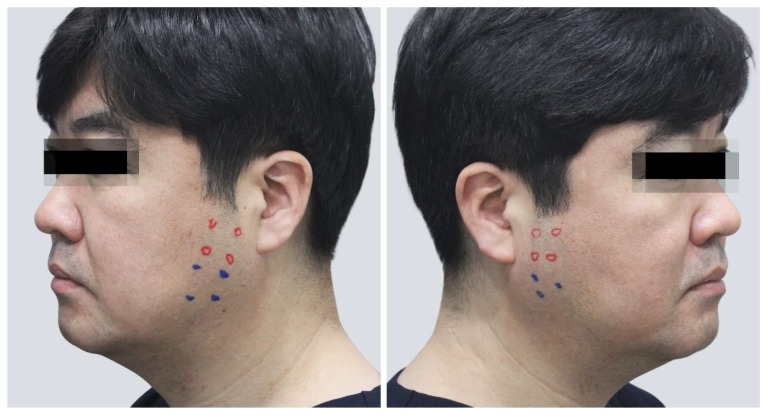
Compensatory hypertrophy due to long-term treatment. The red color points represent the area where excessive symptoms of compensatory hypertrophy appear during clenching; Blue color points represent the injection points where BoNT injections are administered periodically over a long period of time. (Reproduced with the permission of courtesy from Cho HJ, Labella clinic).

**Table 1 toxins-12-00202-t001:** Morphological patterns of the superficial part of the muscle belly and tendon structure of the masseter (*n* = 33).

Superficial Tendon Only	Both Superficial and Deep Tendons			
	Type 1a	Type 1b	Type 1c	Type 2
21.2% (7/33)	24.2% (8/33)	30.3% (10/33)	15.2% (5/33)	9.1% (3/33)

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
