# Peer review of "Anatomical Considerations When Treating Compensatory Hypertrophy of the Upper Part of the Masseter after Long-Term Botulinum Neurotoxin Type A Injections"

_toxins, 2020, doi:10.3390/toxins12030202_

Round 1
Reviewer 1 Report
Title and Introduction: Why did the authors target the compensatory hypertrophy treatment? We could use these data for any masseter muscle BoNT injection. Introduction: Most of the Discussion part should be moved to Introduction. The authors should write what they knew before starting the study in Introduction. Materials and Methods: The authors should write the number of male and female of the cadavers. Materials and Methods: The cadaver age were old. We inject BoNT to young and middle aged patient masseter muscle. The authors should discuss. Materials and Methods: Who decided the pattern of muscle belly and tendon structures? Did the authors refer the previous study? If the pattern is the author original idea, how did they validate? Was an anatomist included in this research? Materials and Methods: How did they decide the pattern of muscle belly and tendon structures? How many researchers did check the pattern? Did they decide with or without discussion? The author should write the detail of the methods.Author Response
Please see the attachment.

Reviewer 2 Report
This study describes anatomical investigations of the upper part of the masseter muscle in a small sample of cadavers from elderly Asian individuals. These studies show data on different types of upper masseter anatomy and tendons, and it is logically inferred that these anatomical differences affect BoNT distribution. However, no studies have actually been conducted with BoNTs to confirm these conclusions, and no literature is cited that has conducted such studies. Please address the following concerns:
- Considering that this manuscript is submitted to the Journal 'Toxins', amazingly little background on the mechanism of BoNT treatment is provided. Please add in the Introduction.
- Please discuss the absence of data directly demonstrating effects of tendons on BoNT diffusion
- Please add more information on the nerve innervating the upper masseter muscle and distribution of nerve endings and the effects of the various tendon structures on nerve distribution. This is important as the BoNT enters the neuronal synapse at the neuromuscular junction.
- Please include innervation into the schematic in Figure 4.
- All Figures: please include labels, please mark each panel with A, B, C, etc, and please include the asterisk mentioned in the legend of Figure 2A.
- Materials and Methods: Was patient (or relatives) consent given?
Round 2
Reviewer 1 Report
The manuscript was improved.
Author Response
Thank you for the valuable advice.
Reviewer 2 Report
no additional comments
Author Response
Thank you for the valuable advice.